# IMPROVED LEARNING OF ONE-HIDDEN-LAYER CONVOLUTIONAL NEURAL NETWORKS WITH OVERLAPS

## ABSTRACT

We propose a new algorithm to learn a one-hidden-layer convolutional neural network where both the convolutional weights and the outputs weights are parameters to be learned. Our algorithm works for a general class of (potentially overlapping) patches, including commonly used structures for computer vision tasks. Our algorithm draws ideas from (1) isotonic regression for learning neural networks and (2) landscape analysis of non-convex matrix factorization problems. We believe these findings may inspire further development in designing provable algorithms for learning neural networks and other complex models. While our focus is theoretical, we also present experiments that illustrate our theoretical findings.

## 1 INTRODUCTION

Giving provably efficient algorithms for learning neural networks is a core challenge in machine learning theory. The case of convolutional architectures has recently attracted much interest due to their many practical applications. Recently Brutzkus & Globerson (2017) showed that distribution-free learning of one simple non-overlapping convolutional filter is NP-hard. A natural open question is whether we can design provably efficient algorithms to learn convolutional neural networks under mild assumptions.

We consider a convolutional neural network of the form

$$f\left(\mathbf{x}, \mathbf{w}, \mathbf{a}\right) = \sum_{j=1}^{k} a_j \sigma\left(\mathbf{w}^\top \mathbf{P}_j \mathbf{x}\right) \tag{1}$$

where $\mathbf{w} \in \mathbb{R}^r$ is a shared convolutional filter, $\mathbf{a} \in \mathbb{R}^k$ is the second linear layer and

$$\mathbf{P}_j = [\underbrace{\mathbf{0}}_{(j-1)s} \underbrace{\mathbf{I}}_{r} \underbrace{\mathbf{0}}_{d-(j-1)s+r}] \in \mathbb{R}^{r \times d}$$

selects the $((j-1)s+1)$-th to $((j-1)s+r)$-th coordinates of $\mathbf{x}$ with stride $s$ and $\sigma\left(\cdot\right)$ is the activation function. Note here that both $\mathbf{w}$ and $\mathbf{a}$ are unknown vectors to be learned and there may be overlapping patches because the stride size $s$ may be smaller than the filter size $r$.

**Our Contributions** We give the *first* efficient algorithm that can provably learn a convolutional neural network with *two* unknown layers with commonly used overlapping patches. Our main result is the following theorem.

**Theorem 1.1** (Main Theorem (Informal)). *Suppose $s \geqslant \lfloor \frac{r}{2} \rfloor + 1$ and the marginal distribution is symmetric and isotropic. Then the convolutional neural network defined in equation 1 with piecewise linear activation functions is learnable in polynomial time.*

We refer readers to Theorem 3.1 for the precise statement.

**Technical Insights** Our algorithm is a novel combination of the algorithm for isotonic regression and the landscape analysis of non-convex problems. First, inspired by recent work on isotonic regression, we extend the idea in Goel et al. (2018) to reduce learning a CNN with piecewise linear activation to learning a convolutional neural network with *linear* activation (c.f. Section 4). Second,

we show learning a linear convolutional filter can be reduced to a non-convex matrix factorization problem which admits a provably efficient algorithm based on non-convex geometry (Ge et al., 2017a). Third, in analyzing our algorithm, we present a robust analysis of Convotron algorithm proposed by Goel et al. (2018), in which we draw connections to the spectral properties of Toeplitz matrices. We believe these ideas may inspire further development in designing provable learning algorithms for neural networks and other complex models.

**Related Work** From the point of view of learning theory, it is well known that training is computational infeasible in the worst case (Goel et al., 2016; Brutzkus & Globerson, 2017). Thus distributional assumptions are needed for efficient learning. A line of research has focused on analyzing the dynamics of gradient descent conditioned on the input distribution being standard Gaussian (Tian, 2017; Soltanolkotabi, 2017; Li & Yuan, 2017; Zhong et al., 2017b; Brutzkus & Globerson, 2017; Zhong et al., 2017a; Du et al., 2017). Specifically for convolutional nets, existing analyses heavily relied on the analytical formulas which can only be derived if the input is Gaussian and patches are non-overlapping.

Recent work has tried to relax the Gaussian input assumption and the non-overlapping structure for learning convolutional filters. Du et al. (2017) showed if the patches are sufficiently close to each other then stochastic gradient descent can recover the true filter. Goel et al. (2018) proposed a modified iterative algorithm inspired from isotonic regression that gives the first recovery guarantees for learning a filter for commonly used overlapping patches under much weaker assumptions on the distribution. However, these two analyses only work for learning one unknown convoutional filter.

Moving away from gradient descent, various works have shown positive results for learning general simple fully connected neural networks in polynomial time and sample complexity under certain assumptions using techniques such as kernel methods (Goel et al., 2016; Zhang et al., 2015; Goel & Klivans, 2017a;b) and tensor decomposition (Sedghi & Anandkumar, 2014; Janzamin et al., 2015). The main drawbacks include the shift to improper learning for kernel methods and the knowledge of the probability density function for tensor methods. In contrast to this, our algorithm is proper and does not assume that the input distribution is known.

Learning a neural network is often formulated as a non-convex problem. If the objective function satisfies (1) all saddle points and local maxima are strict (i.e., there exists a direction with negative curvature), and (2) all local minima are global (no spurious local minmum), then noise-injected (stochastic) gradient descent (Ge et al., 2015; Jin et al., 2017) finds a global minimum in polynomial time. Recent work has studied these properties for the landscape of neural networks (Kawaguchi, 2016; Choromanska et al., 2015; Hardt & Ma, 2016; Haeffele & Vidal, 2015; Mei et al., 2016; Freeman & Bruna, 2016; Safran & Shamir, 2016; Zhou & Feng, 2017; Nguyen & Hein, 2017a;b; Ge et al., 2017b; Zhou & Feng, 2017; Safran & Shamir, 2017; Du & Lee, 2018). A crucial step in our algorithm is reducing the convolutional neural network learning problem to matrix factorization and using the geometric properties of matrix factorization.

## 2 Preliminaries

We use bold-faced letters for vectors and matrices. We use $\|\cdot\|_2$ to denote the Euclidean norm of a finite-dimensional vector. For a matrix $\mathbf{A}$, we use $\lambda_{\max}(\mathbf{A})$ to denote its eigenvalue and $\lambda_{\min}(\mathbf{A})$ its smallest singular value. Let $O(\cdot)$ and $\Omega(\cdot)$ denote standard Big-O and Big-Omega notations, only hiding absolute constants.

In our setting, we have $n$ data points $\{\mathbf{x}_i, y_i\}_{i=1}^n$ where $\mathbf{x}_i \in \mathbb{R}^d$ and $y \in \mathbb{R}$. We assume the label is generated by a two-layer convolutional neural network with filter size $r$, stride $s$ and $k$ hidden neurons. Compactly we can write the formula in the following form: $y_i = f(\mathbf{x}_i, \mathbf{w}^*, \mathbf{a}^*)$, $\mathbf{x}_i \sim \mathcal{Z}$ where the prediction function $f$ is defined in equation 1. To obtain a proper scaling, we let $\|\mathbf{w}^*\|_2 \|\mathbf{a}^*\|_2 = \sigma_1$. We also define the induced patch matrix as

$$\mathbf{P}(\mathbf{x}) = [\mathbf{P}_1\mathbf{x} \quad \ldots \quad \mathbf{P}_k\mathbf{x}] \in \mathbb{R}^{r \times k}$$

which will play an important role in our algorithm design. Our goal is to properly learn this convolutional neural network, i.e., design a polynomial time algorithm which outputs a pair $(\mathbf{w}, \mathbf{a})$ that satisfies $\mathbb{E}_{\mathbf{x} \sim \mathcal{Z}} \left[ (f(\mathbf{w}, \mathbf{a}, \mathbf{x}) - f(\mathbf{w}^*, \mathbf{a}^*, \mathbf{x}))^2 \right] \leqslant \epsilon$.

---

**Algorithm 1** Learning One-hidden-Layer Convolutional Network

---
**Input**: Input distribution $\mathcal{Z}$. Number of iterations: $T_1, T_2$. Number of samples: $T_3$. Step sizes: $\eta_1 > 0, \eta_2 > 0$.
**Output**: Parameters of the one-hidden-layer CNN: $\mathbf{w}$ and $\mathbf{a}$.

1: **Stage 1**: Run Double Convotron (Algorithm 2) for $T_1$ iterations with step size $\eta_1$ to obtain $\mathbf{a}^{(T_1)}$.
2: **Stage 2**: Run Convotron (Algorithm 3) using $\mathbf{a}^{(T_1)}$ and $-\mathbf{a}^{(T_1)}$ for $T_2$ iterations and step size $\eta_2$ to obtain $\mathbf{w}^{(+)}$ and $\mathbf{w}^{(-)}$.
3: **Stage 3**: Choose parameters with lower empirical loss on $T_3$ samples drawn from $\mathcal{Z}$ from $\left(\mathbf{w}^{(+)}, \mathbf{a}^{(T)}\right)$ and $\left(\mathbf{w}^{(-)}, -\mathbf{a}^{(T)}\right)$.

---

# 3 Main Result

In this section we describe our main result. We first list our main assumptions, followed by the detailed description of our algorithm. Lastly we state the main theorem which gives the convergence guarantees of our algorithm.

## 3.1 Assumptions

Our first assumption is on the input distribution $\mathcal{Z}$. We assume the input distribution is symmetric, bounded and has identity covariance. The symmetry assumption is used in Goel et al. (2018) and many learning theory papers Baum (1990). The identity covariance assumption is true if the data whitened. Further, in many architectures, the input of certain layers is assumed having these properties because of the use of batch normalization (Ioffe & Szegedy, 2015) or other techniques. Lastly, the boundedness is a standard regularity assumption to exclude pathological input distributions. We remark that this assumption considerably weaker than the standard Gaussian input distribution assumption used in Tian (2017); Zhong et al. (2017a); Du et al. (2017), which has the rotational invariant property.

**Assumption 3.1** (Input Distribution Assumptions). *We assume the input distribution satisfies the following conditions.*

- *Symmetry:* $\mathbb{P}(\mathbf{x}) = \mathbb{P}(-\mathbf{x})$.

- *Identity covariance:* $\mathbb{E}_{\mathbf{x} \sim \mathcal{Z}}\left[\mathbf{x}\mathbf{x}^T\right] = \mathbf{I}$.

- *Boundedness:* $\forall \mathbf{x} \sim \mathcal{Z}, \|\mathbf{x}\|_2 \leqslant B$ *almost surely for some* $B > 0$.

Our second assumption is on the patch structure. In this paper we assume the stride is larger than half of the filter size. This is indeed true for a wide range of convolutional neural network used in computer vision. For example some architecture has convolutional filter of size 3 and stride 2 and some use non-overlapping architectures (He et al., 2016).

**Assumption 3.2** (Large Stride). $s \geqslant \lfloor \frac{r}{2} \rfloor + 1$.

Next we assume the activation function is piecewise linear. Commonly used activation functions like rectified linear unit (ReLU), Leaky ReLU and linear activation all belong to this class.

**Assumption 3.3** (Piece-wise Linear Activation).

$$\sigma(x) = \begin{cases} x & \text{if } x \geqslant 0 \\ \alpha x & \text{if } x < 0 \end{cases}.$$

## 3.2 Algorithm

Now we are ready to describe our algorithm (see Algorithm 1). The algorithm has three stages, first we learn the outer layer weights upto sign, second we use these fixed outer weights to recover the filter weight and last we choose the best weight combination thus recovered.

**Stage 1: Learning the Non-overlapping Part of the Convolutional Filter and Linear Weights**
Our first observation is even if there may be overlapping patches, as long as there exists some non-overlapping part, we can learn this part and the second layer jointly. To be specific, with filter size being $r$ and stride being $s$, if $s \geqslant \lfloor \frac{r}{2} \rfloor + 1$, for $j = 1. \ldots, k$ we define the selection matrix for the non-overlapping part of each patch

$$\mathbf{P}_j^{non} = [ \underbrace{\mathbf{0}}_{(j-1)s+r} \ \underbrace{\mathbf{I}}_{2s-r} \ \underbrace{\mathbf{0}}_{d-(j+1)s} ] \in \mathbb{R}^{(2s-r) \times d}.$$

Note that for any $j_1 \neq j_2$, there is no overlapping between the selected coordinates by $\mathbf{P}_{j_1}^{non}$ and $\mathbf{P}_{j_2}^{non}$. Therefore, for a filter $\mathbf{w}$, there is a segment $[w_{r-s+1}, \ldots, w_s]$ with length $(2s-r)$ which acts on the non-overlapping part of each patches. We denote $\mathbf{w}_{non} = [w_{r-s+1}, \ldots, w_s]$ and our goal in this stage is to learn $\mathbf{w}_{non}^*$ and $\mathbf{a}^*$ jointly.

In this stage, our algorithm proceeds as follows. Given $\mathbf{w}_{non}, \mathbf{a}$ and a sample $(\mathbf{x}, y)$, we define

$$\mathbf{g}(\mathbf{w}_{non}, \mathbf{a}, \mathbf{x}, y) = \frac{2}{1+\gamma} \left( \hat{f}(\mathbf{w}_{non}, \mathbf{a}, \mathbf{x}) - y \right) \sum_{j=1}^{k} a_i \mathbf{P}_j^{non} \mathbf{x} + \frac{1}{4} \left( \|\mathbf{w}_{non}\|_2^2 - \|\mathbf{a}\|_2^2 \right) \mathbf{w}_{non} \quad (2)$$

$$\mathbf{h}(\mathbf{w}_{non}, \mathbf{a}, \mathbf{x}, y) = \frac{2}{1+\gamma} \left( \hat{f}(\mathbf{w}_{non}, \mathbf{a}, \mathbf{x}) - y \right) \begin{pmatrix} \mathbf{w}_{non}^\top \mathbf{P}_1^{non} \mathbf{x} \\ \ldots \\ \mathbf{w}_{non}^\top \mathbf{P}_k^{non} \mathbf{x} \end{pmatrix} + \frac{1}{4} \left( \|\mathbf{a}\|_2^2 - \|\mathbf{w}_{non}\|_2^2 \right) \mathbf{a} \quad (3)$$

where $\hat{f}(\mathbf{w}_{non}, \mathbf{a}, \mathbf{x}) = \sum_{j=1}^{k} a_j \sigma \left( \mathbf{w}_{non}^\top \mathbf{P}_j^{non} \mathbf{x} \right)$ is the prediction function only using $\mathbf{w}_{non}$.

As will be apparent in Section 4, $\mathbf{g}$ and $\mathbf{h}$ are unbiased estimates of the gradient for the loss function of learning a linear CNN. The term $\frac{1}{4} \left( \|\mathbf{w}_{non}\|_2^2 - \|\mathbf{a}\|_2^2 \right) \mathbf{w}_{non}$ and $\frac{1}{4} \left( \|\mathbf{a}\|_2^2 - \|\mathbf{w}_{non}\|_2^2 \right) \mathbf{a}$ are is the gradient induced by the regularization $\frac{1}{4} \left( \|\mathbf{w}_{non}\|_2^2 - \|\mathbf{a}\|_2^2 \right)^2$, which is used to balance the magnitude between $\mathbf{w}_{non}$ and $\mathbf{a}$ and make the algorithm more stable.

With some initialization $\mathbf{w}_{non}^{(0)}$ and $\mathbf{a}^{(0)}$, we use the following iterative updates inspired by isotonic regression (Goel et al., 2018), for $t = 0, \ldots, T_1 - 1$

$$\mathbf{w}_{non}^{(t+1)} \leftarrow \mathbf{w}_{non}^{(t)} - \eta_1 \mathbf{g}\left( \mathbf{w}_{non}^{(t)}, \mathbf{a}^{(t)}, \mathbf{x}^{(t)}, y^{(t)} \right) + \eta_1 \xi_{\mathbf{w}_{non}}^{(t)}, \quad (4)$$

$$\mathbf{a}^{(t+1)} \leftarrow \mathbf{a}^{(t)} - \eta_1 \mathbf{h}\left( \mathbf{w}_{non}^{(t)}, \mathbf{a}^{(t)}, \mathbf{x}^{(t)}, y^{(t)} \right) + \eta_1 \xi_{\mathbf{a}}^{(t)} \quad (5)$$

where $\eta_1 > 0$ is the step size parameter, $\xi_{\mathbf{w}_{non}}^{(t)}$ and $\xi_{\mathbf{a}}^{(t)}$ are uniformly sampled a unit sphere and at iteration we use a fresh sample $(\mathbf{x}^{(t)}, y^{(t)})$. Here we add isotropic noise $\xi_{\mathbf{w}_{non}}^{(t)}$ and $\xi_{\mathbf{a}}^{(t)}$ because the objective function for learning a linear CNN is non-convex and there may exist saddle points. Adding noise can help escape from these saddle points. We refer readers to Ge et al. (2015) for more technical details regarding this. As will be apparent in Section 4, after sufficient iterations, we obtain a pair $\left( \mathbf{w}^{(T_1)}, \mathbf{a}^{(T_1)} \right)$ such that either it is close to the truth $(\mathbf{w}_{non}^*, \mathbf{a}^*)$ or close to the negative of the truth $(-\mathbf{w}_{non}^*, -\mathbf{a}^*)$.

**Remark 3.1** (Non-overlapping Patches). *If there is no overlap between patches, we can skip Stage 2 because after Stage 1 we have already learned $\mathbf{a}$ and $\mathbf{w}_{non} = \mathbf{w}$.*

**Stage 2: Convotron with fixed Linear Layer** In Stage 1 we have learned a good approximation to the second layer (either $\mathbf{a}^{(T_1)}$ or $-\mathbf{a}^{(T_1)}$). Therefore, the problem reduces to learning a convolutional filter. We run Convotron (Algorithm 3) proposed in Goel et al. (2018) using $\mathbf{a}^{(T_1)}$ and $-\mathbf{a}^{(T_1)}$ to obtain corresponding weight vectors $\mathbf{w}^{(+)}$ and $\mathbf{w}^{(-)}$. We show that the Convotron analysis can be extended to handle approximately known outer layer weight vectors.

**Stage 3: Validation** In stage 2 we have obtained two possible solutions $\left( \mathbf{w}^{(+)}, \mathbf{a}^{(T)} \right)$ and $\left( \mathbf{w}^{(-)}, -\mathbf{a}^{(T)} \right)$. We know at least one of them is close to the ground truth. Closeness in ground truth implies small squared loss (c.f. Lemma A.1). In the last stage we use a validation set to choose

---

**Algorithm 2** Double Convotron

---

Initialize $\mathbf{w}_{non}^{(0)} \in \mathbb{R}^{2s-r}$ and $\mathbf{a}^{(0)} \in \mathbb{R}^k$ randomly
**for** $t = 1$ **to** $T$ **do**
    Draw $(\mathbf{x}^{(t)}, y^{(t)}) \sim \mathcal{Z}$
    Compute $\mathbf{g}\left(\mathbf{w}_{non}, \mathbf{a}, \mathbf{x}^{(t)}, y^{(t)}\right)$ and $\mathbf{h}\left(\mathbf{w}_{non}, \mathbf{a}, \mathbf{x}^{(t)}, y^{(t)}\right)$ according to equation 2 and equation 3.
    Set $\mathbf{w}_{non}^{(t+1)} = \mathbf{w}_{non}^{(t)} - \eta_1 \mathbf{g}\left(\mathbf{w}_{non}^{(t)}, \mathbf{a}^{(t)}, \mathbf{x}^{(t)}, y^{(t)}\right) + \eta_1 \xi_{\mathbf{w}_{non}}^{(t)}$
    Set $\mathbf{a}^{(t+1)} = \mathbf{a}^{(t)} - \eta_1 \mathbf{h}\left(\mathbf{w}_{non}^{(t)}, \mathbf{a}^{(t)}, \mathbf{x}^{(t)}, y^{(t)}\right) + \eta_1 \xi_{\mathbf{a}}^{(t)}$
Return $\mathbf{a}^{(T+1)}$

---

**Algorithm 3** Convotron (Goel et al., 2018)

---

Initialize $\mathbf{w}_1 := 0 \in \mathbb{R}^r$.
**for** $t = 1$ **to** $T$ **do**
    Draw $(\mathbf{x}^{(t)}, y^{(t)}) \sim \mathcal{Z}$
    Let $\mathbf{g}^{(t)} = (y^{(t)} - f(\mathbf{w}^{(t)}, \mathbf{a}, \mathbf{x}^{(t)}))\left(\sum_{i=1}^{k} a_i \mathbf{P}_i \mathbf{x}^{(t)}\right)$
    Set $\mathbf{w}^{(t+1)} = \mathbf{w}^{(t)} + \eta \mathbf{g}^{(t)}$
Return $\mathbf{w}_{T+1}$

---

the right one. To do this, we simply use $T_3 = \text{poly}\left(k, B, \frac{1}{\epsilon}\right)$ fresh samples and output the solution which gives lower squared error.

$$(\mathbf{w}, \mathbf{a}) = \underset{(\mathbf{w}, \mathbf{a}) \in \left\{\left(\mathbf{w}^{(+)}, \mathbf{a}^{(T)}\right), \left(\mathbf{w}^{(-)}, -\mathbf{a}^{(T)}\right)\right\}}{\arg\min} \frac{1}{T_3} \sum_{i=1}^{T_3} \left(y^{(i)} - f\left(\mathbf{w}, \mathbf{a}, \mathbf{x}^{(i)}\right)\right)^2. \tag{6}$$

Since we draw many samples, the empirical estimates will be close to the true loss using standard concentration bounds and choosing the minimum will give us the correct solution.

### 3.3 MAIN THEOREM

The following theorem shows that Algorithm 1 is guaranteed to learn the target convolutinoal neural network in polynomial time. To our knowledge, this is the first polynomial time proper learning algorithm for convolutional neural network with overlapping patches.

**Theorem 3.1** (Theorem 1.1 (Formal))**.** *Under Assumptions 3.1-3.3, if we set* $T_1, T_2, T_3 = \Omega\left(\text{poly}\left(k, B, \frac{1}{\epsilon}\right)\right)$ *and* $\eta_1, \eta_2 = O\left(\text{poly}\left(\frac{1}{k}, \frac{1}{B}, \epsilon\right)\right)$ *then with high probability, Algorithm 1 returns a pair* $(\mathbf{w}, \mathbf{a})$ *which satisfies*

$$\mathbb{E}_{\mathbf{x} \sim \mathcal{Z}}\left[\left(f(\mathbf{w}, \mathbf{a}, \mathbf{x}) - f(\mathbf{w}^*, \mathbf{a}^*, \mathbf{x})\right)^2\right] \leqslant \epsilon.$$

## 4 PROOFS AND TECHNICAL INSIGHTS

In this section we list our key ideas used for designing the Algorithm 1 and proving its correctness. We discuss the analysis stage-wise for ease of understnading.

### 4.1 ANALYSIS OF STAGE 1

**Learning a non-overlapping CNN with linear activation.** We first consider the problem of learning a convolutional neural network with linear activation function and non-overlapping patches. For this setting, we can write the prediction function in a compact form:

$$f(\mathbf{w}, \mathbf{a}, \mathbf{x}) = \mathbf{w}^\top \mathbf{P}(\mathbf{x}) \mathbf{a} = \langle \mathbf{P}(\mathbf{x}), \mathbf{w}\mathbf{a}^\top \rangle.$$

The label also admits this form $y = \langle \mathbf{P}(\mathbf{x}), \mathbf{w}^*(\mathbf{a}^*)^\top \rangle$. A natural way to learn $\mathbf{w}^*$ and $\mathbf{a}^*$ is to consider solving a square loss minimization problem:

$$\min \ell(\mathbf{w}, \mathbf{a}, \mathbf{x}) = \left(\langle \mathbf{P}(\mathbf{x}), \mathbf{w}\mathbf{a}^\top \rangle - \langle \mathbf{P}(\mathbf{x}), \mathbf{w}^*(\mathbf{a}^*)^\top \rangle\right)^2.$$

Now, taking expectation with respect to $\mathbf{x}$, we have

$$L\left(\mathbf{w}, \mathbf{a}\right) = \left\| \mathbf{w}\mathbf{a}^{\top} - \mathbf{w}^{*}\left(\mathbf{a}^{*}\right)^{\top} \right\|_{F}^{2} \tag{7}$$

where the last step we used our assumptions that patches are non-overlapping and the covariance of $\mathbf{x}$ is the identity. From equation 7, it is now apparent that the population $L_2$ loss is just the standard loss for rank-1 matrix factorization problem.

Recent advances in non-convex optimization shows the following regularized loss function

$$L_{reg}\left(\mathbf{w}, \mathbf{a}\right) = \frac{1}{2} \left\| \mathbf{w}\mathbf{a}^{\top} - \mathbf{w}^{*}\left(\mathbf{a}^{*}\right)^{\top} \right\|_{F}^{2} + \frac{1}{8} \left( \|\mathbf{w}\|_{2}^{2} - \|\mathbf{a}\|_{2}^{2} \right)^{2}. \tag{8}$$

satisfies all local minima are global and all saddles points and local maxima has a negative curvature Ge et al. (2017a) and thus allows simple local search algorithm to find a global minimum. Though the objective function in equation 8 is a population risk, we can obtain its stochastic gradient by our samples if we use fresh sample at each iteration. We define

$$\mathbf{g}_{\mathbf{w}}^{(t)} = \left( f\left(\mathbf{w}^{(t)}, \mathbf{a}^{(t)}, \mathbf{x}^{(t)}\right) - y^{(t)} \right) \mathbf{P}\left(\mathbf{x}^{t}\right) \mathbf{a}^{(t)} + \frac{1}{2} \left( \left\| \mathbf{w}^{(t)} \right\|_{2}^{2} - \left\| \mathbf{a}^{t} \right\|_{2}^{2} \right) \mathbf{w}^{(t)} \tag{9}$$

$$\mathbf{g}_{\mathbf{a}}^{(t)} = \left( f\left(\mathbf{w}^{(t)}, \mathbf{a}^{(t)}, \mathbf{x}^{(t)}\right) - y^{(t)} \right) \mathbf{P}\left(\mathbf{x}^{t}\right)^{\top} \mathbf{w}^{(t)} + \frac{1}{2} \left( \left\| \mathbf{a}^{t} \right\|_{2}^{2} - \left\| \mathbf{w}^{(t)} \right\|_{2}^{2} \right) \mathbf{a}^{(t)} \tag{10}$$

where $\left(\mathbf{x}^{(t)}, y^{(t)}\right)$ is the sample we use in the $t$-th iteration. In expectation this is the standard gradient descent algorithm for solving equation 8:

$$\mathbb{E}_{\mathbf{x}}\left[\mathbf{g}_{\mathbf{w}}^{(t)}\right] = \frac{\partial L_{reg}\left(\mathbf{w}^{(t)}, \mathbf{a}^{(t)}\right)}{\partial \mathbf{w}^{(t)}}, \quad \mathbb{E}_{\mathbf{x}}\left[\mathbf{g}_{\mathbf{a}}^{(t)}\right] = \frac{\partial L_{reg}\left(\mathbf{w}^{(t)}, \mathbf{a}^{(t)}\right)}{\partial \mathbf{a}^{(t)}}.$$

With this stochastic gradient oracle at hand, we can implement the noise-injected stochastic gradient descent proposed in Ge et al. (2015).

$$\mathbf{w}^{(t+1)} = \mathbf{w}^{(t)} - \eta \mathbf{g}_{\mathbf{w}}^{(t)} + \eta \xi_{\mathbf{w}}^{(t)}, \quad \mathbf{a}^{(t+1)} = \mathbf{w}^{(t)} - \eta \mathbf{g}_{\mathbf{a}}^{(t)} + \eta \xi_{\mathbf{a}}^{(t)}$$

where $\xi_{\mathbf{w}}^{(t)}$ and $\xi_{\mathbf{a}}^{(t)}$ are sampled from a unit sphere. Theorem 6 in Ge et al. (2015) implies after polynomial iterations, this iterative procedure returns an $\epsilon$-optimal solution of the objective function equation 8 with high probability.

**Learning non-overlapping part of a CNN with piece-wise linear activation function**   Now we consider piece-wise linear activation function. Our main observation is that we can still obtain a stochastic gradient oracle for the *linear* convolutional neural network using equation 2 and equation 3. Formally, we have the following theorem.

**Lemma 4.1** (Properties of Stochastic Gradient for Linear CNN). *Define*

$$L_{reg}\left(\mathbf{w}_{non}, \mathbf{a}\right) = \frac{1}{2} \left\| \mathbf{w}_{non}\mathbf{a}^{\top} - \mathbf{w}_{non}^{*}\left(\mathbf{a}^{*}\right)^{\top} \right\|_{F}^{2} + \frac{1}{8} \left( \|\mathbf{w}_{non}\|_{2}^{2} - \|\mathbf{a}\|_{2}^{2} \right)^{2}.$$

*Under Assumption 3.1, we have*

$$\mathbb{E}_{\mathbf{x}}\left[\mathbf{g}\left(\mathbf{w}_{non}, \mathbf{a}, \mathbf{x}, y\right)\right] = \frac{\partial L_{reg}\left(\mathbf{w}_{non}, \mathbf{a}\right)}{\partial \mathbf{w}_{non}}, \mathbb{E}_{\mathbf{x}}\left[\mathbf{h}\left(\mathbf{w}_{non}, \mathbf{a}, \mathbf{x}, y\right)\right] = \frac{\partial L_{reg}\left(\mathbf{w}_{non}, \mathbf{a}\right)}{\partial \mathbf{a}}$$

*where $\mathbf{g}\left(\mathbf{w}_{non}, \mathbf{a}, \mathbf{x}, y\right)$ and $\mathbf{h}\left(\mathbf{w}_{non}, \mathbf{a}, \mathbf{x}, y\right)$ are defined in equation 2 and equation 3, respectively. Further, if $\|\mathbf{w}_{non}\|_{2} = O(\text{poly}\left(\sigma_{1}\right)), \|\mathbf{a}\|_{2} = O(\text{poly}\left(\sigma_{1}\right))$, then the differences are also bounded*

$$\left\| \mathbf{g}\left(\mathbf{w}_{non}, \mathbf{a}, \mathbf{x}, y\right) - \frac{\partial L_{reg}\left(\mathbf{w}_{non}, \mathbf{a}\right)}{\partial \mathbf{w}_{non}} \right\| \leqslant D, \left\| \mathbf{h}\left(\mathbf{w}_{non}, \mathbf{a}, \mathbf{x}, y\right) - \frac{\partial L_{reg}\left(\mathbf{w}_{non}, \mathbf{a}\right)}{\partial \mathbf{a}} \right\|_{2} \leqslant D$$

*for some $D = O\left(\text{poly}\left(B, k, \sigma_{1}\right)\right)$.*

Here the expectation of $\mathbf{g}$ and $\mathbf{h}$ are equal to the gradient of the objective function for linear CNN because we assume the input distribution is symmetric and the activation function is piece-wise linear. This observation has been stated in Goel et al. (2018) and based on this property, Goel et al.

(2018) proposed Convotron algorithm (Algorithm 3), which we use in our stage 2. Lemma 4.1 is a natural extension of Lemma 2 of Goel et al. (2018) that we show even for one-hidden-layer CNN, we can still obtain an unbiased estimate of the gradient descent for linear CNN.

Now with Lemma 4.1 at hand, we can use the theory from non-convex matrix factorization. Ge et al. (2017a) has shown if $\eta_1 = O\left(\text{poly}\left(\frac{1}{k}, \frac{1}{B}, \epsilon\right)\right)$ then for all iterates, with high probability, $\left\|\mathbf{w}_{non}^{(t)}\right\|_2 = O(\text{poly}(\sigma_1)), \|\mathbf{a}^t\|_2 = O(\text{poly}(\sigma_1))$. Therefore, we can apply the algorithmic result in Ge et al. (2015) and obtain the following convergence theorem.

**Theorem 4.1** (Convergence of Stage 1). *If* $\mathbf{w}^{(0)} = O\left(\sqrt{\sigma_1}\right)$, $\mathbf{a}^{(0)} = O\left(\sqrt{\sigma_1}\right)$, *and* $\eta_1 = O\left(\text{poly}\left(\frac{1}{k}, \frac{1}{B}, \epsilon\right)\right)$ *then after* $T_1 = O\left(\text{poly}\left(r, k, B, \frac{1}{\epsilon}\right)\right)$ *we have*

$$\left\|\frac{\mathbf{a}^{(T_1)}}{\left\|\mathbf{a}^{(T_1)}\right\|_2} - \frac{\mathbf{a}^*}{\|\mathbf{a}^*\|_2}\right\|_2 \leqslant \epsilon. \ or \ \left\|\frac{\mathbf{a}^{(T_1)}}{\left\|\mathbf{a}^{(T_1)}\right\|_2} + \frac{\mathbf{a}^*}{\|\mathbf{a}^*\|_2}\right\|_2 \leqslant \epsilon.$$

### 4.2 ANALYSIS OF STAGE 2

After Stage 1, we have approximately recovered the outer layer weights. We use these as fixed weights and run Convotron to obtain the filter weights. The analysis of Convotron inherently handles average pooling as the outer layer. Here we extend the analysis of Convotron to handle any fixed outer layer weights and also handle noise in these outer layer weights. Formally, we obtain the following theorem:

**Theorem 4.2.** *(Learning the Convolutional Filter) Suppose* $\|\mathbf{a} - \mathbf{a}^*\|_2 \leqslant \epsilon$ *for* $\epsilon \leqslant \frac{1}{k^3\|\mathbf{w}^*\|_2}$ *and without loss of generality[1] let* $\|\mathbf{a}\|_2 = \|\mathbf{a}^*\|_2 = 1$. *For suitably chosen* $\eta = O\left(\text{poly}\left(\frac{1}{k}, \frac{1}{B}\right)\right)$, *Convotron (modified) returns* $\mathbf{w}$ *such that with a constant probability,* $\|\mathbf{w} - \mathbf{w}^*\|_2 \leqslant O(k^3\|\mathbf{w}^*\|\epsilon)$ *in* $\text{poly}(k, \|\mathbf{w}^*\|, B, \log(1/\epsilon))$ *iterations.*

Note that we present the theorem and proof for covariance being identity and no noise in the label but it can be easily extended to handle non-identity convariance with good condition number and bounded (in expectation) probabilistic concept noise.

Our analysis closely follows that from Goel et al. (2018). However, in contrast to the known second layer setting considered in Goel et al. (2018), we only know an approximation to the second layer and a robust analysis is needed. Another difficulty arises from the fact that the convergence rate depends on the least eigenvalue of $\mathbf{P}^{\mathbf{a}} := \sum_{1\leqslant i,j\leqslant k} a_i a_j P_i P_j^T$. By simple algebra, we can show that the matrix has the following form:

$$\mathbf{P}^{\mathbf{a}}(i,j) = \begin{cases} 1 & \text{if } i = j \\ \sum_{i=1}^{k-1} a_i a_{i+1} & \text{if } |i-j| = s \\ 0 & \text{otherwise.} \end{cases}$$

Using property of Toeplitz matrices, we show the least eigenvalue of $\mathbf{P}^{\mathbf{a}}$ is lower bounded by $1 - \cos\left(\frac{\pi}{k+1}\right)$ (c.f. Theorem A.2) for all $\mathbf{a}$ with norm 1.

### 4.3 ANALYSIS OF STAGE 3

Here we show how we can pick the correct hypothesis from the two possible hypothesis. Under our assumptions, the individual loss $(y^{(i)} - f(\mathbf{w}, \mathbf{a}, \mathbf{x}^{(i)}))^2$ is bounded. sThus, a direct application of Hoeffding inequality gives the following guarantee.

**Theorem 4.3.** *Suppose* $T_3 = \Omega\left(\text{poly}\left(r, k, B, \frac{1}{\epsilon}\right)\right)$ *and let* $(\mathbf{w}, \mathbf{a})$. *If either* $\left(\mathbf{w}^{(+)}, \mathbf{a}^{T_1}\right)$ *or* $\left(\mathbf{w}^{(-)}, -\mathbf{a}^{T_1}\right)$ *has population risk smaller than* $2\epsilon$, *then let* $(\mathbf{w}, \mathbf{a})$ *be the output according to equation 6, then with high probability*

$$\mathbb{E}_{\mathbf{x}\sim\mathcal{Z}}\left[f(\mathbf{w}, \mathbf{a}, \mathbf{x} - f(\mathbf{w}^*, \mathbf{a}^*, \mathbf{x}))^2\right] \leqslant \epsilon.$$

---

[1]Note that we can assume that the outer layers have norm 1 by using the normalized weight vectors since the activations are scale invariant.

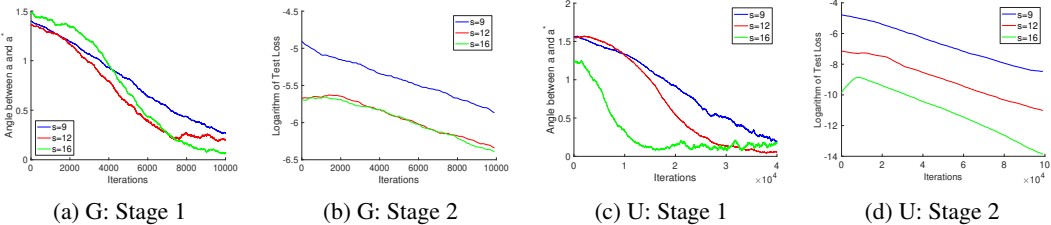

(a) G: Stage 1  (b) G: Stage 2  (c) U: Stage 1  (d) U: Stage 2

Figure 1: Evaluation of Algorithm 1. Gaussian input (G): 1a and 1b. Uniform input (U): 1c and 1d.

### 4.4 PUTTING THINGS TOGETHER: PROOF OF THEOREM 3.1

Now we put our analyses for Stage 1-3 together and prove Theorem 3.1. By Theorem 4.1, we know we have $\mathbf{a}^{(T_1)}$ such that $\left\|\mathbf{a}^{T_1} - \mathbf{a}^*\right\| \leqslant O\left(\frac{\epsilon}{r^{1/2}k^{5/2}\sigma_1}\right)$ (without loss of generality, we assume $\mathbf{a}$ and $\mathbf{a}^*$ are normalized) with $\eta_1 = O\left(\text{poly}\left(\frac{1}{k}, \frac{1}{B}, \frac{1}{\sigma_1}, \epsilon\right)\right)$ and $|S_1| = \text{poly}\left(r, k, B, \sigma_1, \frac{1}{\epsilon}\right)$.

Now with Theorem 4.2, we know with $\eta = O\left(\text{poly}\left(\frac{1}{k}, \frac{1}{B}\right)\right)$ and $|S_2| = O\left(\text{poly}\left(k, \sigma_1, \log\frac{1}{\epsilon}\right)\right)$ we have either $\left\|\mathbf{w}^{(+)} - \mathbf{w}^*\right\|_2 \leqslant \frac{\epsilon}{\sigma_1 r^{1/2}k^{3/2}}$ or $\left\|\mathbf{w}^{(-)} - \mathbf{w}^*\right\|_2 \leqslant \frac{\epsilon}{\sigma_1 r^{1/2}k^{3/2}}$. Lastly, the following lemma bounds the loss of each instance in terms of the closeness of parameters.

**Lemma 4.2.** *For any* $\mathbf{a}$ *and* $\mathbf{w}$, *we have*

$$\left(f\left(\mathbf{w}^*, \mathbf{a}^*, \mathbf{x}\right) - f\left(\mathbf{w}^{(t)}, \mathbf{a}, \mathbf{x}\right)\right)^2 \leqslant 2k\left(\|\mathbf{a}\|_2^2 \|\mathbf{w} - \mathbf{w}^*\|_2^2 + \|\mathbf{a} - \mathbf{a}^*\|_2^2 \|\mathbf{w}^*\|_2^2\right) \|\mathbf{x}\|_2^2.$$

Therefore, we know either $\left(\mathbf{w}^{(+)}, \mathbf{a}^{(T_1)}\right)$ or $\left(\mathbf{w}^{(-)}, -\mathbf{a}^{(T_1)}\right)$ achieves $\epsilon$ prediction error. Now combining Theorem 4.3 and Lemma A.1 we obtain the desired result.

## 5 EXPERIMENTS

In this section we use simulations to verify the effectiveness of our proposed method. We fix input dimension $d = 160$ and filter size $r = 16$ for all experiments and vary the stride size $s = 9, 12, 16$. For all experiments, we generate $\mathbf{w}^*$ and $\mathbf{a}^*$ from a standard Gaussian distribution and use $10,000$ samples to calculate the test error. Note in Stage 2 of Algorithm we need to test $\mathbf{a} = \mathbf{a}^{(T_1)}$ and $-\mathbf{a}^{(T_1)}$. Here we only report the one with better performance in the Stage 2 because in Stage 3 we can decide which one is better. To measure the performance of Stage 1, we use the angle between $\mathbf{a}^t$ an $\mathbf{a}^*$ (in radians). We first test Gaussian input distribution $\mathbf{x} \sim N(\mathbf{0}, \mathbf{I})$. Figure 1a shows the convergence in Stage 1 of Algorithm 1 with $T_1 = 10000$ and $\eta_1 = 0.0001$. Figure 1b shows the convergence in Stage 2 of Algorithm 1 with $T_2 = 10000$ and $\eta_2 = 0.0001$. We then test uniform input distribution $\mathbf{x} \sim \mathbf{Unif}[-\sqrt{3}, \sqrt{3}]^d$ (this distribution has identity covariance). Figure 1c shows the convergence in Stage 1 of Algorithm 1 with $T_1 = 40000$ and $\eta_1 = 0.0001$. Figure 1d shows the convergence in Stage 2 of Algorithm 1 with $T_2 = 100000$ and $\eta_2 = 0.00001$. Note for both input distributions and all choices of stride size, our algorithm achieves low test error..

## 6 CONCLUSION AND FUTURE WORK

In this paper, we propose the first efficient algorithm for learning a one-hidden-layer convolutional neural network with possibly overlapping patches. Our algorithm draws ideas from isotonic regression, landscape analysis of non-convex problem and spectral analysis of Toeplitz matrices. These findings can inspire further development in this field.

Our next step is extend our ideas to design provable algorithms that can learn complicated models consisting of multiple filters. To solve this problem, we believe the recent progress on landscape design (Ge et al., 2017b) may be useful.

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

# A    USEFUL LEMMAS/THEOREMS

In this section we present a few lemmas/theorems that are useful for our analysis.

*Proof of Lemma 4.2.* Observe that,

$$\left(f\left(\mathbf{w}^*, \mathbf{a}^*, \mathbf{x}\right) - f\left(\mathbf{w}^{(t)}, \mathbf{a}, \mathbf{x}\right)\right)^2$$
$$\leqslant 2\left(\left(f\left(\mathbf{w}^*, \mathbf{a}^*, \mathbf{x}\right) - f\left(\mathbf{w}^*, \mathbf{a}, \mathbf{x}\right)\right)^2 + \left(f\left(\mathbf{w}^*, \mathbf{a}, \mathbf{x}\right) - f\left(\mathbf{w}, \mathbf{a}, \mathbf{x}\right)\right)^2\right)$$

since $(a+b)^2 \leqslant 2\left(a^2 + b^2\right)$ for all $a, b \in \mathbb{R}$.

The first term can be bounded as follows,

$$
\begin{aligned}
\left(f\left(\mathbf{w}^*, \mathbf{a}^*, \mathbf{x}\right) - f\left(\mathbf{w}^*, \mathbf{a}, \mathbf{x}\right)\right)^2 &= \left(\sum_{i=1}^{k}\left(a_i^* - a_i\right)\sigma\left(\left(\mathbf{w}^*\right)^T \mathbf{P}_i \mathbf{x}\right)\right)^2 \\
&\leqslant \left(\sum_{i=1}^{k}\left|a_i^* - a_i\right|\left|\left(\mathbf{w}^*\right)^T \mathbf{P}_i \mathbf{x}\right|\right)^2 \\
&\leqslant \left(\sum_{i=1}^{k}\left|a_i^* - a_i\right|\left\|\mathbf{w}^*\right\|_2 \left\|\mathbf{x}\right\|_2\right)^2 \\
&\leqslant k\left\|\mathbf{a}^* - \mathbf{a}\right\|_2^2 \left\|\mathbf{w}^*\right\|_2^2 \left\|\mathbf{x}\right\|_2^2.
\end{aligned}
$$

Here the first inequality follows from observing that $\sigma(a) \leqslant |a|$ and the last follows from $\|\mathbf{v}\|_1 \leqslant \sqrt{k}\|\mathbf{v}\|_2$ for all $\mathbf{v} \in \mathbb{R}^k$.

Similarly, the other term can be bounded as follows,

$$
\begin{aligned}
\left(f\left(\mathbf{w}^*, \mathbf{a}, \mathbf{x}\right) - f\left(\mathbf{w}, \mathbf{a}, \mathbf{x}\right)\right)^2 &= \left(\sum_{i=1}^{k} a_i\left(\sigma\left(\left(\mathbf{w}^*\right)^T \mathbf{P}_i \mathbf{x}\right) - \sigma\left(\mathbf{w}^T \mathbf{P}_i \mathbf{x}\right)\right)\right)^2 \\
&\leqslant \left(\sum_{i=1}^{k}\left|a_i\right|\left|\left(\mathbf{w}^* - \mathbf{w}\right)^T \mathbf{P}_i \mathbf{x}\right|\right)^2 \\
&\leqslant \left(\sum_{i=1}^{k}\left|a_i\right|\left\|\mathbf{w}^* - \mathbf{w}\right\|_2 \left\|\mathbf{x}\right\|_2\right)^2 \\
&\leqslant k\left\|\mathbf{a}\right\|_2^2 \left\|\mathbf{w}^* - \mathbf{w}\right\|_2^2 \left\|\mathbf{x}\right\|_2^2.
\end{aligned}
$$

Here we use the Lipschitz property of $\sigma$ to get the first inequality. The lemma follows from combining the above two.                $\square$

The following lemma extends this to the overall loss.

**Lemma A.1.** *For any* $\mathbf{a}$ *and* $\mathbf{w}$,

$$\mathbb{E}\left[\left(f\left(\mathbf{w}^*, \mathbf{a}^*, \mathbf{x}\right) - f\left(\mathbf{w}^{(t)}, \mathbf{a}, \mathbf{x}\right)\right)^2\right] \leqslant 2kB\left(\left\|\mathbf{a}\right\|_2^2 \left\|\mathbf{w} - \mathbf{w}^*\right\|_2^2 + \left\|\mathbf{a} - \mathbf{a}^*\right\|_2^2 \left\|\mathbf{w}^*\right\|_2^2\right).$$

The following lemma from Goel et al. (2018) is key to our analysis.

**Lemma A.2** (Lemma 1 of Goel et al. (2018)). *For all* $\mathbf{a}, \mathbf{b} \in \mathbb{R}^n$, *if* $\mathcal{Z}$ *is symmetric then,*

$$\mathbb{E}_{\mathbf{x} \sim \mathcal{Z}}\left[\sigma\left(\mathbf{a}^T \mathbf{x}\right)\left(\mathbf{b}^T \mathbf{x}\right)\right] = \frac{1+\alpha}{2}\mathbb{E}_{\mathbf{x} \sim \mathcal{Z}}\left[\left(\mathbf{a}^T \mathbf{x}\right)\left(\mathbf{b}^T \mathbf{x}\right)\right].$$

The following well-known theorem is useful for bounding eigenvalues of matrices.

**Theorem A.1** (Gershgorin Circle Theorem Weisstein (2003)). *For a* $n \times n$ *matrix* $\mathbf{A}$, *define* $R_i := \sum_{j=1, j \neq i}^{n}\left|\mathbf{A}_{i,j}\right|$. *Each eigenvalue of* $\mathbf{A}$ *must lie in at least one of the disks* $\{z : |z - \mathbf{A}_{i,i}| \leqslant R_i\}$.

The following lemma bounds the eigenvalue of the weighted patch matrices.

**Theorem A.2.** *For all* $\mathbf{a} \in \mathbb{S}^{k-1}$,

$$\lambda_{\min}\left(\mathbf{P}^{\mathbf{a}}\right) \geqslant 1 - \cos\left(\frac{\pi}{k+1}\right) \quad and \quad \lambda_{\max}\left(\mathbf{P}^{\mathbf{a}}\right) \leqslant 1 + \cos\left(\frac{\pi}{k+1}\right).$$

*Proof.* Since $s \geqslant \lfloor\frac{r}{2}\rfloor + 1$, only adjacent patches overlap, and it is easy to verify that the matrix $\mathbf{P}^{\mathbf{a}}$ has the following structure:

$$\mathbf{P}^{\mathbf{a}}(i,j) = \begin{cases} 1 & \text{if } i = j \\ \sum_{i=1}^{k-1} a_i a_{i+1} & \text{if } |i - j| = s \\ 0 & \text{otherwise.} \end{cases}$$

Using the Gershgorin Circle Theorem, stated below, we can bound the eigenvalues, $\lambda_{\min}\left(\mathbf{P}^{\mathbf{a}}\right) \geqslant 1 - \left|\sum_{i=1}^{k-1} a_i a_{i+1}\right|$ and $\lambda_{\max}\left(\mathbf{P}^{\mathbf{a}}\right) \leqslant 1 + \left|\sum_{i=1}^{k-1} a_i a_{i+1}\right|$.

To bound the eigenvalues, we will bound $\left|\sum_{i=1}^{k-1} a_i a_{i+1}\right|$ by maximizing it over all $\mathbf{a}$ such that $\|\mathbf{a}\|_2 = 1$. We have $\max\left\{\left|\sum_{i=1}^{k-1} a_i a_{i+1}\right|\right\} = \max\left\{\sum_{i=1}^{k-1} a_i a_{i+1}\right\}$ since the maximum can be achieved by setting all $a_i$ to be non-negative. This can alternatively be viewed as $\max_{\|\mathbf{a}\|_2=1} \mathbf{a}^T \mathbf{M} \mathbf{a} = \lambda_{\max}\left(\mathbf{M}\right)$ where $\mathbf{M}$ is a tridiagonal symmetric Toeplitz matrix as follows:

$$\mathbf{M}(i,j) = \begin{cases} 1/2 & \text{if } |i - j| = 1 \\ 0 & \text{otherwise.} \end{cases}$$

It is well known that the eigenvalues of this matrix are of the form $\cos\left(\frac{i\pi}{k+1}\right)$ for $i = 1, \ldots, k$ (c.f. Boeóttcher & Grudsky (2005)). The maximum eigenvalue is thus $\cos\left(\frac{\pi}{k+1}\right)$. This gives us the result. □

# B    OMITTED PROOFS

## B.1    PROOF OF LEMMA 4.1

First by definition, we have

$$\mathbb{E}_{\mathbf{x}}\left[\mathbf{g}\left(\mathbf{w}_{non}, \mathbf{a}, \mathbf{x}, y\right)\right] = \mathbb{E}_{\mathbf{x}}\left[\frac{2}{1+\gamma}\left(\hat{f}\left(\mathbf{w}_{non}, \mathbf{a}, \mathbf{x}\right) - y\right)\sum_{j=1}^{k} a_j \mathbf{P}_j^{non}\mathbf{x} + \frac{1}{4}\left(\|\mathbf{w}_{non}\|_2^2 - \|\mathbf{a}\|_2^2\right)\mathbf{w}_{non}\right].$$

Because the input distribution is symmetric and the covariance is identity, by Lemma A.2, we have

$$\mathbb{E}_{\mathbf{x}}\left[\frac{2}{1+\gamma}\hat{f}\left(\mathbf{w}_{non}, \mathbf{a}, \mathbf{x}\right)\sum_{j=1}^{k} a_j \mathbf{P}_j^{non}\mathbf{x}\right] = \mathbb{E}_{\mathbf{x}}\left[\frac{2}{1+\gamma}\sum_{j=1}^{k} a_j \sigma\left(\mathbf{w}_{non}^\top \mathbf{P}_j^{non}\mathbf{x}\right)\sum_{j=1}^{k} a_j \mathbf{P}_j^{non}\mathbf{x}\right]$$

$$= \mathbb{E}_{\mathbf{x}}\left[\sum_{j=1}^{k} a_j \mathbf{w}_{non}^\top \mathbf{P}_j^{non}\mathbf{x}\sum_{j=1}^{k} a_j \mathbf{P}_j^{non}\mathbf{x}\right]$$

$$= \|\mathbf{a}\|_2^2 \mathbf{w}_{non}.$$

Similarly, we have

$$\mathbb{E}_{\mathbf{x}}\left[y\sum_{j=1}^{k} a_j \mathbf{P}_j^{non}\mathbf{x}\right] = \mathbf{a}^\top \mathbf{a}^* \mathbf{w}^*.$$

Also recall

$$\frac{\partial L_{reg}\left(\mathbf{w}, \mathbf{a}\right)}{\partial \mathbf{w}} = \|\mathbf{a}\|_2^2 \mathbf{w}_{non} - \mathbf{a}^\top \mathbf{a}^* \mathbf{w}^* + \frac{1}{4}\left(\|\mathbf{w}_{non}\|_2^2 - \|\mathbf{a}\|_2^2\right)\mathbf{w}_{non}.$$

Thus

$$\mathbb{E}_{\mathbf{x}}\left[\mathbf{g}\left(\mathbf{w}_{non}, \mathbf{a}, \mathbf{x}, y\right)\right] = \frac{\partial L_{reg}\left(\mathbf{w}, \mathbf{a}\right)}{\partial \mathbf{w}}.$$

The proof for $\mathbf{h}\left(\mathbf{w}_{non}, \mathbf{a}, \mathbf{x}, y\right)$ is similar.

To obtain a bound of the gradient, note that

$$\left\|\sum_{j=1}^{k} a_j \sigma\left(\mathbf{w}_{non}^{\top} \mathbf{P}_j^{non} \mathbf{x}\right) \sum_{j=1}^{k} a_j \mathbf{P}_j^{non} \mathbf{x}\right\| \leqslant \sum_{j=1}^{k}|a_j|\left|\mathbf{w}_{non}^{\top} \mathbf{P}_j^{non} \mathbf{x}\right|\left\|\sum_{j=1}^{k} a_j \mathbf{P}_j^{non} \mathbf{x}\right\|$$

$$\leqslant \max_{j}|a_j| \sum_{j=1}^{k}\|\mathbf{w}_{non}\|_2\left\|\mathbf{P}_j^{non}\mathbf{x}\right\|_2 \cdot \|\mathbf{a}\|_2 \|\mathbf{x}\|_2$$

$$\leqslant k\|\mathbf{a}\|_2^2\|\mathbf{x}\|_2^2\|\mathbf{w}\|_2$$

$$=\text{poly}\left(k, \sigma_1, B\right).$$

Similar argument applies to $y\sum_{j=1}^{k} a_j \mathbf{P}_j^{non}\mathbf{x}$.

## B.2   Proof of Theorem 4.2

We follow the Convotron analysis and include the changes. Define $S_t = \{(\mathbf{x}_1, y_1), \ldots, (\mathbf{x}_t, y_t)\}$. The modified gradient update is as follows,

$$\mathbf{g}^{(t)} = \left(y_t - f\left(\mathbf{w}^{(t)}, \mathbf{a}, \mathbf{x}_t\right)\right)\left(\sum_{i=1}^{k} a_i P_i \mathbf{x}_t\right)$$

The dynamics of Convotron can then be expressed as follows:

$$\mathbb{E}_{\mathbf{x}_t, y_t}\left[\left\|\mathbf{w}^{(t)} - \mathbf{w}^*\right\|_2^2 - \left\|\mathbf{w}^{(t+1)} - \mathbf{w}^*\right\|_2^2 | S_{t-1}\right] = 2\eta\mathbb{E}_{\mathbf{x}_t, y_t}\left[\left(\mathbf{w}^* - \mathbf{w}^{(t)}\right)^T \mathbf{g}^{(t)} | S_{t-1}\right] - \eta^2\mathbb{E}_{\mathbf{x}_t, y_t}\left[||\mathbf{g}^{(t)}||^2 | S_{t-1}\right].$$

We have,

$$\mathbb{E}_{\mathbf{x}_t, y_t}\left[\left(\mathbf{w}^* - \mathbf{w}^{(t)}\right)^T \mathbf{g}^{(t)} | S_{t-1}\right]$$

$$= \mathbb{E}_{\mathbf{x}_t, y_t}\left[\left(\mathbf{w}^* - \mathbf{w}^{(t)}\right)^T \left(y_t - f\left(\mathbf{w}^{(t)}, \mathbf{a}, \mathbf{x}_t\right)\right)\left(\sum_{i=1}^{k} a_i P_i \mathbf{x}_t\right) | S_{t-1}\right]$$

$$= \mathbb{E}_{\mathbf{x}_t}\left[\left(\mathbf{w}^* - \mathbf{w}^{(t)}\right)^T \left(f\left(\mathbf{w}^*, \mathbf{a}^*, \mathbf{x}_t\right) - f\left(\mathbf{w}^{(t)}, \mathbf{a}, \mathbf{x}_t\right)\right)\left(\sum_{i=1}^{k} a_i P_i \mathbf{x}_t\right) | S_{t-1}\right]$$

$$= \sum_{1\leqslant i,j\leqslant k} \mathbb{E}_{\mathbf{x}_t}\left[\left(a_i^*\sigma\left((\mathbf{w}^*)^T P_i\mathbf{x}_t\right) - a_i\sigma\left(\left(\mathbf{w}^{(t)}\right)^T P_i\mathbf{x}_t\right)\right)\left(a_j(\mathbf{w}^*)^T - a_j\left(\mathbf{w}^{(t)}\right)^T\right) P_j\mathbf{x}_t | S_{t-1}\right]$$

$$= \frac{1+\alpha}{2} \sum_{1\leqslant i,j\leqslant k} \mathbb{E}_{\mathbf{x}_t}\left[\left(\left(a_i^*(\mathbf{w}^*)^T - a_i\left(\mathbf{w}^{(t)}\right)^T\right) P_i\mathbf{x}_t\right)\left(a_j(\mathbf{w}^*)^T - a_j\left(\mathbf{w}^{(t)}\right)^T\right) P_j\mathbf{x}_t | S_{t-1}\right]$$

$$\tag{11}$$

$$= \frac{1+\alpha}{2} \sum_{1\leqslant i,j\leqslant k}\left(a_i^*(\mathbf{w}^*)^T - a_i\left(\mathbf{w}^{(t)}\right)^T\right) P_i\mathbb{E}_{\mathbf{x}_t}[\mathbf{x}_t\mathbf{x}_t^T]P_j^T\left(a_j\mathbf{w}^* - a_j\mathbf{w}^{(t)}\right) \tag{12}$$

$$= \frac{1+\alpha}{2} \sum_{1\leqslant i,j\leqslant k}\left(a_i^*(\mathbf{w}^*)^T - a_i(\mathbf{w}^*)^T + a_i(\mathbf{w}^*)^T - a_i\left(\mathbf{w}^{(t)}\right)^T\right) P_iP_j^T\left(a_j\mathbf{w}^* - a_j\mathbf{w}^{(t)}\right)$$

$$= \frac{1+\alpha}{2}\left(\left((\mathbf{w}^*)^T - (\mathbf{w}^*)^T\right)\mathbf{P^a}(\mathbf{w}^* - \mathbf{w}^*) + \sum_{1\leqslant i,j\leqslant k}(a_i^* - a_i) a_j(\mathbf{w}^*)^T P_iP_j^T\left(\mathbf{w}^* - \mathbf{w}^{(t)}\right)\right)$$

$$\tag{13}$$

$$\geqslant \frac{1+\alpha}{2} \left( \lambda_{\min}\left(\mathbf{P^a}\right) \left\|\mathbf{w}^{(t)} - \mathbf{w}^*\right\|_2^2 - \left\|\mathbf{w}^*\right\|_2 \left\|\sum_{1 \leqslant i \leqslant k} \left(a_i^* - a_i\right) P_i\right\|_2 \left\|\sum_{1 \leqslant j \leqslant k} a_j P_j\right\|_2 \left\|\mathbf{w}^{(t)} - \mathbf{w}^*\right\|_2 \right)$$

(14)

$$\geqslant \frac{1+\alpha}{2} \left( \lambda_{\min}\left(\mathbf{P^a}\right) \left\|\mathbf{w}^{(t)} - \mathbf{w}^*\right\|_2^2 - k \left\|\mathbf{w}^*\right\| \left\|\mathbf{a}^* - \mathbf{a}\right\|_2 \left\|\mathbf{a}\right\|_2 \left\|\mathbf{w}^{(t)} - \mathbf{w}^*\right\|_2 \right)$$

$$\geqslant \frac{1+\alpha}{2} \left( \lambda_{\min}\left(\mathbf{P^a}\right) \left\|\mathbf{w}^{(t)} - \mathbf{w}^*\right\|_2^2 - k\epsilon \left\|\mathbf{w}^*\right\|_2 \left\|\mathbf{w}^{(t)} - \mathbf{w}^*\right\|_2 \right)$$

(11) follows from using Lemma A.2, (13) follows from defining $\mathbf{P^a} := \sum_{1 \leqslant i,j \leqslant k} a_i a_j P_i P_j^T$, (12) follows from setting the covariance matrix to be identity and (14) follows from observing that $\mathbf{P^a}$ is symmetric, thus $\forall \mathbf{x}, \mathbf{x}^T \mathbf{P^a} \mathbf{x} \geqslant \lambda_{\min}\left(\mathbf{P^a}\right) \|\mathbf{x}\|_2^2$ as well as lower bounding the second term in terms of the norms of the corresponding parts.

Now we bound the variance of $\mathbf{g}^{(t)}$.

$$\mathbb{E}_{\mathbf{x}_t, y_t}[||\mathbf{g}^{(t)}||^2 | S_{t-1}]$$

$$= \mathbb{E}_{\mathbf{x}_t, y_t} \left[ \left(y_t - f\left(\mathbf{w}^{(t)}, \mathbf{a}, \mathbf{x}_t\right)\right)^2 \left\|\sum_{i=1}^k a_i P_i \mathbf{x}_t\right\|^2 \bigg| S_{t-1} \right]$$

$$\leqslant \lambda_{\max}\left(\mathbf{P^a}\right) \mathbb{E}_{\mathbf{x}_t} \left[ \left(f\left(\mathbf{w}^*, \mathbf{a}^*, \mathbf{x}_t\right) - f\left(\mathbf{w}^{(t)}, \mathbf{a}, \mathbf{x}_t\right)\right)^2 ||\mathbf{x}_t||^2 \bigg| S_{t-1} \right]$$

(15)

$$\leqslant 2k \lambda_{\max}\left(\mathbf{P^a}\right) \left( \|\mathbf{a}\|_2^2 \left\|\mathbf{w}^{(t)} - \mathbf{w}^*\right\|_2^2 + \|\mathbf{a} - \mathbf{a}^*\|_2^2 \|\mathbf{w}^*\|_2^2 \right) \mathbb{E}_{\mathbf{x}_t} \left[ \|\mathbf{x}_t\|_2^4 \right]$$

(16)

$$\leqslant 2kB \lambda_{\max}\left(\mathbf{P^a}\right) \left( \left\|\mathbf{w}^{(t)} - \mathbf{w}^*\right\|_2^2 + \epsilon^2 \|\mathbf{w}^*\|_2^2 \right)$$

(17)

(15) follows from observing that $\left\|\sum_{i=1}^k a_i \mathbf{P}_i \mathbf{x}\right\|^2 \leqslant \lambda_{\max}\left(\mathbf{P^a}\right) \|\mathbf{x}\|_2^2$ for all $\mathbf{x}$ and (16) follows from Lemma 4.2.

Combining the above equations and taking expectation over $S_{t-1}$, we get

$$\mathbb{E}_{S_t}\left[\left\|\mathbf{w}^{(t+1)} - \mathbf{w}^*\right\|_2^2\right] \leqslant \left(1 - 2\eta\beta + \eta^2\gamma\right) \mathbb{E}_{S_{t-1}}\left[\left\|\mathbf{w}^{(t)} - \mathbf{w}^*\right\|_2^2\right] + 2\eta\alpha\epsilon \mathbb{E}_{S_{t-1}}\left[\left\|\mathbf{w}^{(t)} - \mathbf{w}^*\right\|_2\right] + \eta^2\chi\epsilon^2$$

$$\leqslant \left(1 - 2\eta\beta + \eta^2\gamma\right) \mathbb{E}_{S_{t-1}}\left[\left\|\mathbf{w}^{(t)} - \mathbf{w}^*\right\|_2^2\right] + 2\eta\delta\epsilon \sqrt{\mathbb{E}_{S_{t-1}}\left[\left\|\mathbf{w}^{(t)} - \mathbf{w}^*\right\|_2^2\right]} + \eta^2\chi\epsilon^2$$

for $\beta = \frac{1+\alpha}{2} \lambda_{\min}\left(\mathbf{P^a}\right)$, $\gamma = 2\lambda_{\max}\left(\mathbf{P^a}\right)kB$, $\delta = \frac{1+\alpha}{2}k\|\mathbf{w}^*\|_2$ and $\chi = 2\lambda_{\max}\left(\mathbf{P^a}\right)kB\|\mathbf{w}^*\|_2^2$.

From Theorem A.2, we have that $\lambda_{\min}\left(\mathbf{P^a}\right) = 1 - \cos\left(\frac{\pi}{k+1}\right) = \Omega\left(1/k^2\right)$ (by Taylor expansion) implying $\beta = \omega\left(1/k^2\right)$ and $\gamma = O\left(kB\right)$, $\chi = O\left(kB\|\mathbf{w}^*\|_2^2\right)$.

We set $\eta = \beta \min\left(\frac{1}{\gamma}, \frac{1}{\chi}\right)$. First we show that $\mathbb{E}_{S_{t-1}}[\|\mathbf{w}^{(t)} - \mathbf{w}^*\|_2^2] \leqslant 1$ for all iterations $t$. We prove this inductively. For $t = 1$, since $w_1 = 0$, this is satisfied. Let us assume it holds for iteration $t$, then we have that,

$$\mathbb{E}_{S_t}\left[\left\|\mathbf{w}^{(t+1)} - \mathbf{w}^*\right\|_2^2\right] \leqslant \left(1 - 2\eta\beta + \eta^2\gamma\right) \mathbb{E}_{S_{t-1}}\left[\left\|\mathbf{w}^{(t)} - \mathbf{w}^*\right\|_2^2\right] + 2\eta\delta\epsilon \sqrt{\mathbb{E}_{S_{t-1}}\left[\left\|\mathbf{w}^{(t)} - \mathbf{w}^*\right\|_2^2\right]} + \eta^2\chi\epsilon^2$$

$$\leqslant 1 - 2\eta\beta + \eta^2\gamma + 2\eta\delta\epsilon + \eta^2\chi\epsilon^2$$

$$\leqslant 1 - 2\eta\beta + \eta\beta + 2\eta\delta\epsilon + \eta\beta\epsilon$$

$$\leqslant 1 - \eta\left(\beta - (\delta + \beta)\epsilon\right) \leqslant 1$$

The last inequality follows from $\epsilon \leqslant \frac{1}{k^3 \|\mathbf{w}^*\|_2} \leqslant \frac{\beta}{\delta + \beta}$. Thus we have that for each iteration, $\mathbb{E}_{S_{t-1}}[\|\mathbf{w}^{(t)} - \mathbf{w}^*\|_2^2] \leqslant 1$. Substituting this in the recurrence and solving the recurrence gives

us,

$$\mathbb{E}_{S_t}[\left\|\mathbf{w}^{(t+1)} - \mathbf{w}^*\right\|_2^2] \leqslant \left(1 - 2\eta\beta + \eta^2\gamma\right) \mathbb{E}_{S_{t-1}}[\left\|\mathbf{w}^{(t)} - \mathbf{w}^*\right\|_2^2] + 2\eta\delta\epsilon + \eta^2\chi\epsilon^2$$

$$\leqslant \left(1 - \eta\beta\right) \mathbb{E}_{S_{t-1}}[\left\|\mathbf{w}^{(t)} - \mathbf{w}^*\right\|_2^2] + 2\eta\delta\epsilon + \eta\beta\epsilon^2$$

$$\leqslant \left(1 - \eta\beta\right)^t \left\|\mathbf{w}_1 - \mathbf{w}^*\right\|_2^2 + \left(2\eta\delta\epsilon + \eta\beta\epsilon^2\right) \sum_{i=0}^{t-1} \left(1 - \eta\beta\right)^i$$

$$\leqslant \left(1 - \eta\beta\right)^t + \frac{2\delta\epsilon}{\beta} + \epsilon^2$$

Thus for $T = O\left(\frac{1}{\eta\beta} \log\left(\frac{1}{\epsilon}\right)\right)$, we have,

$$\mathbb{E}_{S_t}[\left\|\mathbf{w}^{(t+1)} - \mathbf{w}^*\right\|_2^2] \leqslant O\left(\frac{\delta\epsilon}{\beta}\right) = O\left(k^3 \left\|\mathbf{w}^*\right\|_2 \epsilon\right).$$

Now using Markov's inequality, we know that the above holds for some constant probability.

### B.3 PROOF OF THEOREM 4.3

For $i = T_2 + 1, \ldots, T_3$, define $z^{(i)} = \left(y^{(i)} - f\left(\mathbf{w}, \mathbf{a}, \mathbf{x}^{(i)}\right)\right)^2$. Using our assumptions, we know $z^{(i)} \leqslant O\left(\text{poly}\left(r, k, B\right)\right)$ almost surely. Now applying Hoeffding inequality we obtain our desired result.

