# OpenReview forum: "Improved Learning of One-hidden-layer Convolutional Neural Networks with Overlaps"
_ICLR.cc/2019/Conference_

### Official Review · AnonReviewer2 · 2018-11-02

**Rating:** 6
**Confidence:** 4

**Review:**

This paper gives a new algorithm for learning a two layer neural network which involves a single convolutional filter and a weight vector for different locations. The algorithm works on any symmetric input data. The techniques in this paper combines two previous approaches: 1. the algorithm Convotron for learning a single convolutional filter (while the second layer has fixed weight) on any symmetric input distributions; 2. non-convex optimization for low rank matrix factorization.

The main observation in the paper is that if the overlap in the convolutions is not large (in the sense that each location of the convolution has at least one input coordinate that is not used in any other locations), then the weight that corresponds to the non-overlapping part and the weights in the second layer can be computed by a matrix factorization step (the paper gives a way to estimate a gradient that is similar to the gradient for a linear neural network, and then the problem is very similar to a rank-1 matrix factorization). After this step, we know the second layer and the algorithm can generalize the previous Convotron algorithm to learn the full convolutional filter.

This is an interesting observation that allows the algorithm to learn a two-layer neural network. On the other hand this two layer neural network is still a bit limited as there is still only one convolutional filter, and in particular there is only one local and global optimum (up to scaling the two layers). The observation also limited how much the patches can overlap which was not a problem in the original convotron algorithm.

Overall I feel the paper is interesting but a bit incremental.

---

> ### Author Response · Authors · 2018-11-16
> **Response**
>
> Thanks for your feedback. We would like to emphasize the following points.
>
> First, the generalization of Convotron [1] to handle approximately known weights in the second layer is highly non-trivial. It requires an analysis of the new patch matrix that is weighted by the coefficients. We use a substantially different technique from [1] to show the given property. In particular, we use properties of Toeplitz matrices which might be of independent interest.
>
> Second, prior work on learning the same architecture cannot handle any overlap, and in practice, less than half overlap is often used (patch size 3X3 with stride 2).
>
> Third, our main goal of this paper to handle a broader class of convolutional neural networks that have 1) overlaps and 2) more than one layers. For multiple filters, to our knowledge, even the simplest case, the problem of learning a fully connected neural network has not been resolved yet. Nevertheless, once that problem is solved,  one can combine our approach to learn convolutional neural networks with multiple filters.
>
> [1] Surbhi Goel, Adam Klivans, and Raghu Meka. Learning one convolutional layer with overlapping patches. arXiv preprint arXiv:1802.02547, 2018.

---

### Official Review · AnonReviewer1 · 2018-11-03
**This is a theoretical paper investigating learning a one-hidden-layer CNN with overlap**

**Rating:** 5
**Confidence:** 1

**Review:**

I believe the authors need to give more intuition on the importance of such a study, and how it can lead to improvement in real life application.
The work seems interesting but is limited and as the authors mentioned it might be a good start for further investigation. However, what I really wanted to see was a simple comparison on a dataset like MNIST with conventional CNN being trained via SGD, for example.
Also, there are some small typos you may need to fix, e.g "will be play" -> "will be playing".

---

> ### Author Response · Authors · 2018-11-16
> **Response**
>
> We thank the reviewer for the comments.
> First, we want to emphasize that this is a theory paper and the primary goal of this is paper is to broaden our knowledge on the learnability of convolutional neural networks. Prior work on learning this model via SGD/GD required stronger assumptions such as Gaussian input and no-overlap in the patches. Our techniques help us give strong guarantees under significantly weaker assumptions.
>
> Practically, we consider a simple layer by layer training which can be heuristically extended for deeper layers. For experiments, in this paper, we are specifically looking at the regression problem unlike the classification problem in MNIST. We do present experiments to show the validity of our approach on a synthetic dataset.
>
> We have fixed the typos that you mentioned. Thanks!

---

### Official Review · AnonReviewer4 · 2018-11-18
**Interesting theoretical study of One-hidden-layer Conv Nets**

**Rating:** 6
**Confidence:** 3

**Review:**

This paper studies the theoretical learning of one-hidden-layer convolutional neural nets. The main result is a learning algorithm and provable guarantees using the algorithm.  This result extends previous analysis to handle the learning of the output layer weights, and holds for symmetric input distributions with identity covariance matrix.

At a high level, the proof works by using the non-overlapping part of the filter to reduce the problem to matrix factorization.
The reduced problem corresponds to learning a rank-one matrix, from which one can learn the output layer weight vector approximately. Given the output weight vector, then the hidden layer weight is learnt using the Convotron algorithm from previous analysis. I think that the technical contribution is interesting.

Weakness: Given the existing work (Goel et al. 2018), I am concerned that the current work is a bit incremental. Secondly, it is unclear if the technical insight has any applications or not. How does the proposed algorithm work on real world data? Even some simple comparisons to other algorithms on a few datasets would provide insight.

Question: Where does Assumption 3.2 arise in the proof? Is it necessary (for the proof)?

Other issues: A few typos you may need to fix (e.g. the S notation in Thm 3.1, first sentence in Sec 4.3).

---

> ### Author Response · Authors · 2018-11-20
> **Response**
>
> We thank for your thoughtful review.
>
> First, we want to emphasize that the generalization of Convotron [1] to handle approximately known weights in the second layer is highly non-trivial. It requires an analysis of the new patch matrix that is weighted by the coefficients. We use a substantially different technique from [1] to show the given property. In particular, we use properties of Toeplitz matrices which might be of independent interest.
>
> Second, we believe our theoretically inspired layer by layer training procedure can be heuristically extended for deeper layers.
>
> Regarding your question on Assumption:
> 1. First, it guarantees that there is a non-overlapping part of the filter. In the first stage of our algorithm, we first learn this non-overlapping part and the second layer jointly.
> 2. In the second stage, using Assumption 3.2 we show P^{a} (page 7) has a lower bounded least eigenvalue.
>
> Thank you for pointing out the typos, we have fixed them.

---

### Meta-Review · Area_Chair1 · 2018-12-19

**Recommendation:** Reject
**Confidence:** 5

**Metareview:**

The reviewers seem to reach a consensus that the contribution of the paper is somewhat incremental give the prior work of Goel et al and that a main drawback of the paper is that it's not clear the similar technique can be applied to multiple **convolutional filters**. The authors mentioned in the response that some of the techniques can be heuristically applied to multiple layers, but the AC is skeptical about it because, with multiple layers and multiple convolutional filters, one has to deal with the permutation invariance caused by the multiple convolutional filters. (It's unclear to the AC how one could have a meaningful setting with multiple layers but a single convolution filters.)